# ESD Ideas: Exoplanet, origins of life and biosphere researchers offer a perspective fundamental to ensuring humanity's future

Daniel Duzdevich[1], Arwen E. Nicholson[2], Raphaëlle D. Haywood[2]

[1] Department of Chemistry, The University of Chicago, Chicago, IL 60637, USA
[2] Department of Physics and Astronomy and Global Systems Institute, University of Exeter, Exeter, EX4 4QL, UK

*Correspondence to*: Raphaëlle D. Haywood (r.d.haywood@exeter.ac.uk)

**Introduction.** We are at a crucial moment in human history. Earth's natural systems are in extreme flux and threaten the health of civilisation (Romanello et al., 2022). Continued human-induced climate and ecosystem disruption (IPCC, 2023) demonstrate that human power structures are built to function as though they are distinct from Nature. No one would set their house on fire unless they believed themselves immune to the flames; and yet we are setting our house on fire. The collective response by the political, economic and social powers that have precipitated these crises over the next 10-15 years will determine humanity's future over the next millennium. Acknowledging the deep resilience and interdependence of the Earth-life system is a necessary step to mapping a sustainable future. Scientists exploring the histories of planets and life are uniquely positioned to communicate a perspective that is fundamental to our survival: *humanity is wholly embedded in the Earth and its biosphere. There is no escaping our planet and its history.* Only policies that build on this perspective will contribute to a flourishing future for humanity. Here, we offer a few brief glimpses of this cosmic perspective, and call on our colleagues to acknowledge the powerful stories emanating from their work.

## Life is deeply embedded in Earth history

How life began remains one of the biggest unanswered questions in science. There is no fossil record of life's emergence on Earth: weathering, plate tectonics, and the carbon cycle collude to obliterate the imprints of biology as time passes. Nevertheless, we know more about our planetary history than even a few decades ago. We are learning more about other planets both in the solar system and beyond, and this allows us to contextualize Earth and its history. For example, we can now consider the prevalence of terrestrial, temperate planets, the relationship between how planetary systems form and evolve and how likely they are to be hospitable to life, and the influence of the host star on planetary habitability (e.g. Kopparapu et al., 2018; Meadows et al., 2020; Bryson et al., 2021). We also have the example of a rich and thriving biology on our planet. The great achievement of modern biology is the insight that all living things are essentially the same at the molecular level, having evolved from common primordial ancestors. The Last Universal Common Ancestor (LUCA) would have possessed the fundamental properties we associate with living cells (Koonin, 2003). Our primary strategy for experimentally studying the origin of life in the laboratory is the understanding that however life started, its earliest features allowed for a pathway that

lead to LUCA. Research into the origin of life thus points to an ancient event, not long after the birth of our planet (Pearce et al., 2018), that transformed a chemical system into that of a durable biology.

**Acknowledging our place in the biosphere and in the Earth system**

Everything we know in our daily lives, and all the biology teeming around us, has spread from the emergence of life on our
planet billions of years ago. Our relatedness is profound: life is one family sharing one home. Biology has covered the planet very thoroughly: from rock deep beneath the ocean floor to the high reaches of the atmosphere (and even on space probes!), every niche that can support life—and even some that would seem inhospitable—is inhabited (Rothschild, 2001). The biosphere as a whole has shown striking resilience throughout Earth's history, adjusting to cataclysmic changes in its planetary environment over time (Corsetti, Olcott, & Bakermans, 2006). Humans are latecomers to this story. Our very existence is
enabled by the expansive diversity of life that preceded us and that is actively maintaining the niche we occupy. We are inextricably embedded in something much bigger than our senses can register. If we want to ensure a sustainable future for humanity, then we must first acknowledge our place in the biosphere's planetary playground and epic history. *We do not argue that this acknowledgement will lead to a sustainable future, but rather that a sustainable future is impossible without it.* Life has become integrated into Earth's planetary-scale workings over billions of years. Biology, oceans, atmosphere, and land
interact with one another through massive feedback mechanisms. The result is a robust homeostasis: the Earth system maintains an equilibrium and is resilient to small, slow perturbations (Lovelock & Margulis, 1974; Lovelock, 2016). For example, the carbon cycle functions as our planet's thermostat and stabilizes Earth's climate over millions of years (Isson et al., 2020). Life, as a whole, further increases resilience to climate perturbations and thus actively contributes to maintaining habitable temperatures on Earth's surface (Schwartzman & Volk, 1989; Vicca et al., 2022). Our planet and biology are interlaced. All
the living things that make up the biosphere right now are products of an equally long stretch of evolution: from bacteria to conscious beings, each species has evolved to fit a particular environment. Humans evolved to fit a narrow niche, but, in the grand scheme of things, we are a part of the planet in the same way as every other organism. Planet and Life co-evolve together (Schwartzman, Jorgensen, & Fath, 2008): we are not apart from or above this system we call Nature. *Humans will not survive without a robust and resilient biosphere: it is our life support system.*

**Life shapes Earth's astronomical identity**

Our planet, only one among billions of others in the galaxy, is alive. Its biosignatures are visible at astronomical distances (Schwieterman et al., 2018). What about life on planets beyond our solar system? We are now able to estimate exoplanetary composition and equilibrium temperature (NAS, 2018). For some of the closest and largest exoplanets, we can examine their atmospheres and identify constituent gases, which may serve as remotely detectable biosignatures. If life is ubiquitous, we will
detect it on terrestrial, temperate planets in the next decades (Seager, 2014; NAS, 2019).

**Astronomical context is essential for building a sustainable future**

Although we occupy a privileged and exciting moment in our understanding of the universe, *we will not find another haven to escape our planetary and humanitarian crises.* The search for exoplanets and life is about gaining knowledge and perspective about the planet we live on. For example, we are using observations of planets in our solar system and beyond to test and validate climate models (e.g. Pierrehumbert, 2011; see further works in the review by Shields, 2019). Observations of exoplanets and the planets of the solar system (e.g. Meadows et al., 2020) show us what different possible planetary histories and futures may look like (e.g. Nicholson et al., 2018). Although Venus is Earth's sister, it has taken a very different path, to a runaway greenhouse. Examining other planets informs models of planetary evolution with direct observations (e.g. Raymond, Izidoro & Morbidelli, 2020). Building a statistical sample of planetary snapshots will provide insights into the mechanisms that determine stable climate states (Scharf, 2014). Our models of the Earth system, informed by the data of new frontiers in astronomy, could and should help guide policy decisions, serving as civilization's compass for navigating rapid planetary changes.

**Calling our fellow scientists to inspire and lead**

We are explorers of the origin and fate of life and planets. Our research community offers a unique and powerful perspective on humanity's most acute intergenerational challenges: humans are a part of the biosphere, embedded in the Earth system, one among billions of planets in the galaxy (e.g. Grinspoon, 2016; Frank, 2019). This perspective confers immense responsibility. Today's urgent planetary crises demand that we scientists acknowledge and communicate the broader implications of our research (Green, 2020). Our academic freedom gives us unique agency and flexibility to pursue research avenues that align with our values. We exhort our colleagues to move beyond the myth that scientific research is value-free. Social sciences provide clear evidence that science is not value-neutral (e.g. Oreskes, 2019; Sinatra & Hofer, 2021). Engaging in advocacy does not diminish public trust in scientists (Kotcher et al., 2017); on the contrary, the public expects climate advocacy from us (Cologna et al., 2021). Scientists are the most trusted professionals alongside doctors (Ipsos, 2022). The COVID-19 pandemic boosted people's confidence in scientists (Wellcome, 2020), suggesting that society may increasingly turn to us as planetary crises worsen. Here, we have sketched some expansive and immediately relevant perspectives offered by our fields of research, without being prescriptive. Our goal is to encourage our colleagues to be innovative and creative in connecting their research with global crises, while leveraging the trust, credibility, and privilege of the scientific enterprise. We have a duty to not only highlight the cosmic perspective, but to ensure that it serves as a basis for action. Let us educate and inspire, inform and act.

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

**Acknowledgements:**

We are grateful to Prof. Tim Lenton for his insightful comments and advice, to Dr Malcolm Rogge whose editing greatly improved the clarity and impact of this piece, and to Dr Sebastiaan Krijt who generously lent us many fascinating books. We
thank Dr Caleb Scharf and Prof. Carsten Herrmann-Pillath for their constructive reviews of our manuscript. DD is supported in part by a grant from the National Aeronautics and Space Administration (80NSSC22K0188). AEN is a Postdoctoral Fellow supported by a Leverhulme Trust Research Grant (RPG-2020-82). RDH is an Ernest Rutherford Fellow funded by the UK Science and Technology Facilities Council (ST/V004735/1). The authors have no conflicts of interest to disclose.