# Peer review of "ESD Ideas: Exoplanet, origins of life and biosphere researchers offer a perspective fundamental to ensuring humanity's future"

_EGUsphere, 2023_

## Author Response (AR1)

Overall Response: We thank the reviewers for their thoughtful comments. We have made revisions to address specific issues, and added an introductory paragraph to address general concerns. Our piece is now more clear about its target audience and aims, and we have also changed the title to reflect this. All our edits are coloured in red or blue in the attached Word doc.

**General comments:**

This preprint is a short opinion and/or commentary on the relevance of the Earth Sciences, as well as evolutionary and astrobiological sciences, for driving a perspective that is described as critical for guiding human decisions that will determine our future longevity and the future of the Earth's systems. It is a call for colleagues to harness this perspective and to connect their research to broader existential problems. In this sense it is not a research contribution per se.

Comment: While it offers a commendable 'call-to-arms' it presents a very limited argument in support of its primary contention that there is an entrenched conception of humans as distinct from Nature which hinders our most impactful decisions.

Response: See below ("Line 9-10").

Comment: It is mostly a collection of several talking points (life's singular common ancestry, a Gaian planetary system, exoplanets and climates and life elsewhere) that are suggested as enabling a 'compass' for policy decisions and a narrative for guidance in global socio-economic issues.

Response: We have added connecting sentences throughout to more explicitly tie these points together and tighten the narrative of our piece.

See below for our response on the issue of policy.

Comment: The main weakness is that while all of this makes sense, it is not a new vision, and little specific evidence is presented to show how scientific knowledge about (for example) Earth's deep history or the history of other worlds has tangible impact on seeing future directions for humanity.

Response: We argue that a push for the widespread acknowledgement of this perspective in the scientific community is novel. See below in our response to the second reviewer ("Indeed, similar opinions have been raised many times ..."). We are not suggesting tangible solutions, but rather a perspective to anchor future directions for humanity.

Comment: Are there examples where, for instance, predictive climate models have been significantly upgraded with such knowledge? How does the common ancestry of life on Earth inform us about biosphere function? What exactly might future exoplanet data tell us about choices for planetary environmental stability? A little more specificity throughout could help tighten the argument.

Response: See below for our response on the issue of prescribing specific policies or actions. We have also added a few more references to works that use exoplanet/solar-system data to inform our understanding of Earth. We have also added more references to prior writing on this vision, namely Frank (2018) and Grinspoon (2016).

**Specific comments:**

Comment: Line 9-10: "An entrenched…": This is an important statement, but the article does not provide that counterpoint perspective (humans distinct from Nature) so that the reader is left to infer that the various statements about life's history and evolution and planetary processes negate that perspective. i.e. a problem is indicated but then not explicitly stated in the body of the article.

Response: The very reality of environmental crises unambiguously shows that humanity is acting as though it were distinct from Nature. A simple analogy is that no human would intentionally set their home on fire unless they believed they didn't need it. Planet Earth is humanity's only home. We recognize that we had merely implied this, and have now explicitly stated this reasoning to serve as a counterpoint perspective to our own.

Comment 16-17: "…partly because…": This may be true but as stated is ambiguous: is it referring to improved knowledge about planet formation or alternate climate states, or something else? It is a statement that 'feels right' but without an elaboration on the basis is rather superficial.

Response: We thank the reviewer for pointing out this ambiguity. We were referring to the fact that with the discovery of exoplanets, and the relatively new deeper knowledge of solar system bodies, we can now contextualize the properties of our planet relative to others. We have other examples to compare it to, which is a jump forward in our self-awareness as a planetary species. We have eliminated the "partly because" language and added an explanatory sentence followed by examples of major questions that can now be seriously considered.

"For example, we can now consider the prevalence of Earth-like planets, the relationship between how planetary systems form and evolve and how likely they are to be hospitable to life, and the importance of host-star type."

Comment: 19-20: "possessed the fundamental": This is a hypothesis or definition rather than a proven fact, so perhaps soften this statement - if a LUCA existed, as seems plausible, then it would have possessed...etc.

Response: LUCA by definition possessed the fundamental features of extant biology. We have changed the wording to "would have possessed."

Comment: 21: "harbored the potential": What does this mean? As written it suggests a pre-instantiated potential or pattern, whereas most research would suggest that the systems of the first 'life' had the capacity to lead to the emergence or generation of further novelty.

Response: The entities / systems that resulted in biology must have had the potential to result in biology. This does not imply pre-instantiated potential, merely emergent potential. We have changed the language to eliminate any confusion.

"Our primary strategy for experimentally studying the origin of life in the laboratory is the understanding that however life started, its earliest features allowed for a pathway that lead to LUCA."

Comment: 32-33: "If we want to ensure…": It's a plausible sounding statement - but it's also a bit empty of evidence, it is a hypothesis that seems reasonable but what exactly is it about this acknowledgement that would lead to a sustainable future? Can some hint of that or specific examples be given?

Response: This is the same issue as raised in the first comment above (Line 9-10). It is useless pushing for any movement on climate change, for example, if the majority of power structures do not acknowledge we belong to the same system within which climate change operates. Most scientists would presumably acknowledge that humans operate within the Earth-biosphere system, and yet we see almost no mainstream scientific effort to raise this issue among colleagues or with the public, let alone hammer it home. We argue that this failure is fatal because it is a *logical and obvious prerequisite* for wholesale action. The reality of these crises, right now, in our present moment, indicates that humanity as a whole does not acknowledge its intimate connection to the planet or the biosphere. This is a general argument, and we are seeking to make a point that does not require any specific policy positions.

*"We do not argue that this acknowledgement will lead to a sustainable future, but rather that a sustainable future is impossible without it."*

Comment: 42: "the products": This is perhaps rather oversimplifying things, or at least missing the aspect of evolution that is perfectly fine erasing and reinventing functional forms and mechanisms. All species alive today exist as a snapshot within a long and constantly shifting landscape of evolution - should be careful not to suggest that everything today is somehow a special or 'ultimate' outcome.

Response: We agree and are keen to avoid any such misreading. We like the reviewer's language, and would like to incorporate it, with their permission.

"All species alive today are merely a snapshot within a long and constantly shifting landscape of evolution [...]"

Comment: 45-46: "Humans would not survive…": Perhaps it is important to make this statement, but it reads as so self-evident as to not be profound at all.

Response: Emphasizing that we are inextricably embedded in an ancient biosphere is central to our piece. We have changed the wording from "life on Earth" to "a robust and resilient biosphere".

Comment: 51: "identify constituent gases": Clarify that these constituent gases may serve as remotely detectable biosignatures.

Response: We have made this clarification.

Comment: 56-57: "test and validate…": This is partially correct: we can use these observations to test the validity of these models in conditions that deviate from those of the modern Earth - checking whether the models capture climate processes in more 'extreme' or varied conditions than represented here.

Response: We have replaced "test and validate models of Earth's climate" with "test and validate climate models."

Comment: 62: "can guide policy decisions…": The optimism is appreciated but is there a way to state this that is a little more nuanced - "could and should" rather than "can"?

Response: We have made the change, and it now reads: "could and should help guide policy decisions..."

**Technical corrections:**

Comment: Line 20: "primary tool" – suggest "primary strategy"

Response: We agree that this phrasing is more precise, and have incorporated the change.

Comment: 22-23: "chemistry into a durable biology" - suggest "transformed one chemical system into another that we associate with biology"

Response: We appreciate the clarity of the suggestion, but still want to emphasize the durability of the "chemistry we associate with biology": it is a self-propagating and continuously out-of-equilibrium chemistry that has persisted for billions of years. We have changed the phrase to "transformed one chemical system into that of a durable biology."

Comment: 32: "our instincts" – is this the right facility? Senses? Perceptions?

Response: We had not considered this subtle point. "Senses" best describes what we mean.

Comment: 36: "The result is homeostasis:" Perhaps more accurate to say something like "robust homeostasis" since homeostasis could exist in the absence of life (e.g. carbon-silicate cycle can form in an abiotic form).

Response: We agree, and this also fits with our broader arguments.

Comment: 43-44: "…we are a part of the planet…": Sentence seems repetitive from previous statements.

Response: This sentence drives home our primary point.

Comment: 58: typo around exoplanets

Response: "(exo-)planets" invokes both exoplanets and planets of the solar system in one word. We have simply stated what we mean explicitly and written "exoplanets and the planets of the solar system."
* * *
Comment: After reading Caleb Scharf's comment, I can focus on two additional aspects as I fully endorse this comment.

Response: We addressed all of Caleb Scharf's comments above.

Comment: Indeed, similar opinions have been raised many times, so the question is what the authors might add as new perspectives. My first comment is on juxtaposing biosphere and humans. This gives sort of romantic and backward-looking picture of the Earth system which humans have transformed into a 'hybrid planet' (Frank et al., 2017).

Response: We do not juxtapose biosphere and humans. We argue that they are inextricable, and that humans are a part of the biosphere, not separate from it. Our new perspective is that origin of life, exoplanet, and Earth Systems Science research highlight that humans are enmeshed in the Earth's biosphere. Scientists engaged in these disciplines should use this perspective to contextualize environmental crises. A science-informed perspective that we are embedded in Nature is a powerful argument against self-harm: against harming Nature and therefore ourselves.

Comment: There is now a rich literature on the technosphere as newly emerging regulatory sphere of the Earth system (Donges et al., 2017). How should humans locate in this complicated relationship, which, after all, is one if not *the* defining feature of the Anthropocene?

Response: Our piece is not concerned with the technosphere, which we consider one of many ways humans affect the planetary environment. We locate entirely inside the biosphere. We argue that there is no escaping our planet and its history.

Comment: Just arguing that we are in control anyway, and hence simply including the technosphere into the human domain, is certainly wrong (Haff, 2014).

Response: We are not in control any more or less than any other form of life. We are a part of the Earth-life system, and therefore our actions affect it. The technosphere is a component of the biosphere domain. It is wholly a feature of biology on this planet. When humans go extinct, the technosphere will disappear (though its effects on the planet may persist, as have the effects of many previous forms of life).

Comment: The technosphere follows its own evolutionary trajectory. There are many ways how humans can design co-evolutionary regulatory mechanisms, such as in specific context as the recently propagated 'nature-based solutions' (Herrmann-Pillath et al., 2022). I think the opinion piece needs to add more concrete references to such topics which would allow to demonstrate practical consequences of the suggested change of perspectives for policies.

Response: We are not suggesting policies in this piece. Policies are not useful if scientific research about planetary crises is generally ignored, as is now the case. We argue that the perspective itself has value. We are calling our colleagues to acknowledge that this perspective emanates from their work, this understanding that humans are not beyond Nature and should therefore not exploit it. We do not want to be prescriptive because we invoke many disciplines with distinct intellectual traditions, and because the list of policies that flows from this perspective is vast, politically complex (see below), and well beyond the scope of this piece. Here, we aim to alert our colleagues to this connection between their work and environmental crises.

Comment: The second comment continues with pointing to the rich literature in the humanities dealing with 'nature'. For example, environmental philosopher Vogel has radically deconstructed nature and widens the notion to include artefacts with higher systemic complexity, with matches with the previous comment (Vogel, 2015). Juxtaposing nature and humans reinstates the Western epistemologies of dividing subject and object.

Response: We are not reinstating the Western epistemologies of dividing subject and object. We are not making a division. Dominant power structures are making this division. *Based on strictly scientific arguments*, they shouldn't be. We welcome the future prospect of amplifying this message across disciplines insofar as other intellectual traditions point to the same conclusion. Our focus here is based on our expertise. Research about the history of life on Earth and the Earth-life system, and the astronomical perspective offered by exoplanet science all indicate that humans and the Earth-planet system are not distinct. This is our primary point.

Comment: I cannot map this rich debate here (Braidotti, 2019), but just highlight one, which is inspired by a lifelong study of and engagement with native Australians, Povinelli's concept of 'geontopower' (Povinelli, 2016). Such contributions reveal the fallacies of much of the Anthropocene debates among scientists: They overlook that we should not talk about 'humans' in general, but about those humans that were and still are responsible for the tragedy that we face. In other words, there is a deeply political dimension of the issues, related to questions such as whether and how we must radically change our economic

system. Without facing such political realities, calls for arms (as Scharf uses the term) don't know the enemy. Addressing 'humans' can even dilute responsibilities and factually protect the vested interests of the current system.

Response: We agree with the fact that not every human is equally responsible for planetary crises. We are making a perspective statement: all humans are part of Nature, not separate from it. That's a different category of statement from one that assesses responsibility. However, our target audience is scientists. Scientists in general, and certainly scientists in the Global North and especially the West, are acutely responsible for these crises in complicated ways. Our opinion piece is implicitly political: we argue that scientists have a responsibility to use their privileged position and influence to consider and act on this perspective.

---

## Referee Report (RR1)

Thus is a new version of a paper that I commented on previously. I returned to my comment and notice that the authors did not consider my arguments at all. Therefore, I just copy and paste my previous comment below.

I add one specific observation. The authors write: "The result is a robust homeostasis: the Earth system maintains an equilibrium and is resilient to small, slow perturbations." This confuses the concepts of homeostasis and equilibrium. The key point of the Gaia hypothesis is that the Earth maintains homeostasis in *disequilibrium*. This disequilibrium is seen as one reliable indicator of life on other planets. In the 'hybrid planet' framework that I refer to in my comment below, the idea is that in principle, the technosphere can also achieve a similar state of homeostasis while further leveraging the thermodynamic productivity on Earth. On this mechanism, see the blog post by Axel Kleidon: https://technosphere.blog/2019/05/03/do-humans-have-free-will-or-are-our-actions-merely-manifestations-of-a-thermodynamic-imperative-or-are-both-views-right-in-their-own-ways/. This is the physical foundation for the arguments below.

Here is my previous comment.

My first comment is on juxtaposing biosphere and humans. This gives sort of romantic and backward-looking picture of the Earth system which humans have transformed into a 'hybrid planet' (Frank et al., 2017). There is now a rich literature on the technosphere as newly emerging regulatory sphere of the Earth system (Donges et al., 2017). How should humans locate in this complicated relationship, which, after all, is one if not *the* defining feature of the Anthropocene? Just arguing that we are in control anyway, and hence simply including the technosphere into the human domain, is certainly wrong (Haff, 2014). The technosphere follows its own evolutionary trajectory. There are many ways how humans can design co-evolutionary regulatory mechanisms, such as in specific context as the recently propagated 'nature-based solutions' (Herrmann-Pillath et al., 2022). I think the opinion piece needs to add more concrete references to such topics which would allow to demonstrate practical consequences of the suggested change of perspectives for policies.

The second comment continues with pointing to the rich literature in the humanities dealing with 'nature'. For example, environmental philosopher Vogel has radically deconstructed nature and widens the notion to include artefacts with higher systemic complexity, with matches with the previous comment (Vogel, 2015). Juxtaposing nature and humans reinstates the Western epistemologies of dividing subject and object. I cannot map this rich debate here (Braidotti, 2019), but just highlight one, which is inspired by a lifelong study of and engagement with native Australians, Povinelli's concept of 'geontopower' (Povinelli, 2016). Such contributions reveal the fallacies of much of the Anthropocene debates among scientists: They overlook that we should not talk about 'humans' in general, but about those humans that were and still are responsible for the tragedy that we face. In other words, there is a deeply political dimension of the issues, related to questions such as whether and how we must radically change our economic system. Without facing such political realities, calls for arms (as Scharf uses the term) don't know the enemy. Addressing 'humans' can even dilute responsibilities and factually protect the vested interests of the current system.

Braidotti, R., 2019. Posthuman knowledge. Polity, Medford, MA.

Donges, J.F., Lucht, W., Müller-Hansen, F., Steffen, W., 2017. The technosphere in Earth System analysis: A coevolutionary perspective. The Anthropocene Review 4, 23–33. https://doi.org/10.1177/2053019616676608

Frank, A., Kleidon, A., Alberti, M., 2017. Earth as a Hybrid Planet: The Anthropocene in an Evolutionary Astrobiological Context. Anthropocene 19, 13–21. https://doi.org/10.1016/j.ancene.2017.08.002

Haff, P., 2014. Humans and technology in the Anthropocene: Six rules. The Anthropocene Review 1, 126–136. https://doi.org/10.1177/2053019614530575

Herrmann-Pillath, C., Hiedanpää, J., Soini, K., 2022. The co-evolutionary approach to nature-based solutions: A conceptual framework. Nature-Based Solutions 2, 100011. https://doi.org/10.1016/j.nbsj.2022.100011

Povinelli, E.A., 2016. Geontologies: a requiem to late liberalism. Duke University Press, Durham.

Vogel, S., 2015. Thinking like a mall: environmental philosophy after the end of nature. MIT Press, Cambridge, Massachusetts.

---

## Author Response (AR2)

This is a new version of a paper that I commented on previously. I returned to my comment and notice that the authors did not consider my arguments at all. Therefore, I just copy and paste my previous comment below.

The scope of our piece is solely to provide a perspective based on scientific observation- and experiment-driven research in exoplanets, origin of life, and Earth-systems science. Our target audience is other scientists in our fields of research. Our opinion piece is not prescriptive and does not exclude other schools of thought. It is not ontological. We are aware of the substantial literature on these topics from other disciplines, but given the constraints of the format, this is a short opinion piece, not a literature review (we are already at the word limit).

- Please note that we had changed the title in the previous revision from "To address planetary crises, we must understand our place on Earth" to "Exoplanet, origins of life and biosphere researchers offer a perspective fundamental to ensuring humanity's future". This new title unambiguously sets out the scope and goal of our piece.
- We have adjusted the text to address the reviewer's additional comment about homeostasis, as indicated below.
- Furthermore, we direct the reviewer and the editor to our previous response, which addressed the reviewer's comments point-by-point.

I add one specific observation. The authors write: "The result is a robust homeostasis: the Earth system maintains an equilibrium and is resilient to small, slow perturbations." This confuses the concepts of homeostasis and equilibrium.

We appreciate the distinction being made between "equilibrium" and "stable disequilibrium." To avoid confusion, we have edited the above sentence to read "The result is a robust homeostasis: the Earth system is stable and resilient to small, slow perturbations [...]"

The key point of the Gaia hypothesis is that the Earth maintains homeostasis in disequilibrium. This disequilibrium is seen as one reliable indicator of life on other planets. In the 'hybrid planet' framework that I refer to in my comment below, the idea is that in principle, the technosphere can also achieve a similar state of homeostasis while further leveraging the thermodynamic productivity on Earth.

We thank the reviewer for highlighting this additional aspect. We make no comment on the technosphere, and this is beyond the scope of the authors' respective areas of expertise.

On this mechanism, see the blog post by Axel Kleidon: https://technosphere.blog/2019/05/03/do-humans-have-free-will-or-are-our-actions-merely-manifestations-of-a-thermodynamic-imperative-or-are-both-views-right-in-their-own-ways/. This is the physical foundation for the arguments below.

This short opinion piece makes no presumption of attempting to address the incredibly challenging problem of free will. We have, as outlined above, clarified the scope of our piece.

Here is my previous comment.
My first comment is on juxtaposing biosphere and humans. This gives sort of romantic and backward-looking picture of the Earth system which humans have transformed into a 'hybrid planet' (Frank et al., 2017).

We place humans inside the biosphere. Humans are biology, and a part of the planet-life system. The central argument of our piece is *not* to juxtapose biosphere and humans.
We state this explicitly throughout the piece. See for example in the abstract (in italics): "[...] *humanity is wholly embedded in the Earth and its biosphere. There is no escaping our planet and its history.* Only policies that build on this perspective will contribute to a flourishing future for humanity." Further, we are explicit in not promoting specific polices, and are rather here interested in stating that policies grounded in the fallacy that humanity and Nature are distinct are bound to fail because they ignore scientific reality.

There is now a rich literature on the technosphere as newly emerging regulatory sphere of the Earth system (Donges et al., 2017). How should humans locate in this complicated relationship, which, after all, is one if not the defining feature of the Anthropocene? Just arguing that we are in control anyway, and hence simply including the technosphere into the human domain, is certainly wrong (Haff, 2014).

We make no comment on the technosphere, and this is beyond the scope of the authors' respective areas of expertise. We are providing a science-based perspective, which doesn't exclude other schools of thought. Our critique comes directly from the work of scientists that is used to construct these ontological frameworks. The word "Anthropocene" does not appear. We make no argument that "we are in control anyway".

The technosphere follows its own evolutionary trajectory. There are many ways how humans can design co-evolutionary regulatory mechanisms, such as in specific context as the recently propagated 'nature-based solutions' (Herrmann-Pillath et al., 2022). I think the opinion piece needs to add more concrete references to such topics which would allow to demonstrate practical consequences of the suggested change of perspectives for policies.

As stated above, our short piece does not mention the technosphere. Further, we are keen not to be prescriptive, and seek merely to highlight to scientific colleagues that their observation- and experiment-driven research in exoplanets, origin of life, and Earth-systems science places humans fully inside the Earth-biosphere system.

The second comment continues with pointing to the rich literature in the humanities dealing with 'nature'. For example, environmental philosopher Vogel has radically deconstructed nature and widens the notion to include artefacts with higher systemic complexity, with matches with the previous comment (Vogel, 2015). Juxtaposing nature and humans reinstates

the Western epistemologies of dividing subject and object. I cannot map this rich debate here (Braidotti, 2019), but just highlight one, which is inspired by a lifelong study of and engagement with native Australians, Povinelli's concept of 'geontopower' (Povinelli, 2016).

Our piece specifically seeks to bring in a perspective from exoplanet, origins of life and Earth-system science researchers. We are not attempting an extensive literature review.

Such contributions reveal the fallacies of much of the Anthropocene debates among scientists: They overlook that we should not talk about 'humans' in general, but about those humans that were and still are responsible for the tragedy that we face. In other words, there is a deeply political dimension of the issues, related to questions such as whether and how we must radically change our economic system. Without facing such political realities, calls for arms (as Scharf uses the term) don't know the enemy. Addressing 'humans' can even dilute responsibilities and factually protect the vested interests of the current system.

We wholeheartedly agree with the fact that not every human is equally responsible for current planetary crises. We are making a perspective statement: all humans are part of Nature, not separate from it. That's a different category of statement from one that assesses responsibility. Our target audience is scientists. Scientists in general, and certainly scientists in the Global North and especially the West, are acutely responsible for these crises in complicated ways (that cannot be addressed here). Our opinion piece is therefore implicitly political: we argue that scientists have a responsibility to use their privileged position and influence to consider and act on this perspective.

---

## Author Response (AR3)

Dear authors,

Many thanks for your revised manuscript. I would be happy to accept this subject to minor revisions. Below, please find a list of these. Once you have addressed them all, please submit the further revised version. I will then review this, and make a final decision on publication. I am optimistic this means we have a rapid route to acceptance.

We thank the Editor for their thoughtful feedback and encouraging response. We have addressed each point as indicated below.

1)

"Only policies that build on this perspective will contribute to a flourishing future for humanity. Here, we offer a few brief glimpses of this cosmic perspective, and call on our colleagues to acknowledge the powerful stories emanating from their work." I suggest the language is changed to highlight the argument or perhaps advocacy you are developing. For example "Here we argue that these perspectives should inform policies that seek to build a flourishing future for humanity. To that end, we offer a few brief glimpses of this cosmic perspective, and call on our colleagues to acknowledge the powerful stories emanating from their work."

We agree that this shift to highlight advocacy potential may be a more effective presentation for our target audience. We have changed these sentences to read: "Here we offer a few brief glimpses of this cosmic perspective, and argue that it should inform policies that seek to build a flourishing future for humanity. We call on our colleagues to acknowledge the powerful stories emanating from their work."

2)

It's a minor point, and probably a quibble, but "every niche that can support life—and even some that would seem inhospitable—is inhabited (Rothschild, 2001)". Does it make sense to define a niche that is not inhabited? Does using the word niche here risk making the statement tautologous? I think the point you are making is that there is extraordinary diversity in life, that evolution has produce adapations that have allowed life to survive and thrive in even surprising places (e.g. extremophiles).

We agree that technically an unoccupied niche may not qualify as a niche (though perhaps one could formulate the concept of a "potential niche"). We have rephrased the sentence to read, "Biology has covered the planet very thoroughly with an extraordinary diversity of life: from rock deep beneath the ocean floor to the high reaches of the atmosphere (and even on space probes!), every environment that can support life—including many that would seem inhospitable—is inhabited (Rothschild, 2001)."

3)

"Humans will not survive without a robust/ and resilient biosphere: it is our life support system." Yes, but this makes me ask about scope. Is it human extinction that is the risk? Or is it the destruction of complex/industrialised societies? Or is it very large increases in morbidity and mortality? The first is expansive, the last perhaps more immediate? The reason I ask - and will expand later, is that technological-based solutions to such questions can sometimes emphasise

the former. For example, the central (only significant?) risk of longtermism is Homo sapiens extinction.

This is an important point. There is necessarily a selfish (human-centered) element to any discussion about mitigating the effects of the climate crisis. To use the metaphor of the introduction, if we (collectively or individually) did not mind burning to death in our own house, then there would be no reason to do anything. At the opposite end of the spectrum is the argument that humans are completely irrelevant. This leads either to arguments that we should not harm the environment (which is also our human habitat) for the sake of the environment itself, or that because the biosphere is resilient and humans are irrelevant, it does not matter if climate change destroys us and a significant proportion of the environments that are important to us. Our position implicitly rejects both extremes, the strictly human-centered and the strictly non-human-centered, because it places humans within the biosphere, rather than at some aloof position. We also implicitly reject the extreme interpretation of longtermism (or indeed moving the human species off-planet, see below) because neither is in fact a scientifically grounded or viable possibility. We therefore argue that both for the sake of biosphere and for the sake of human survival in the near future, we must accept and acknowledge the place of the latter inside the former. Ultimately, the argument is simple: disrupt the biosphere, disrupt human life. The scope of biosphere disruption will determine the scope of disruption to human life. We find it difficult to expand on this given the length limitations, however, we have appended, "[...] and the scale of disruption to the biosphere will determine the scale of disruption to human life."

4)
"Although we occupy a privileged and exciting moment in our understanding of the universe, we will not find another haven to escape our planetary and humanitarian crises. There is no planet B (Nicholson & Haywood, 2023)." I understand that argument. Perhaps it could be spelt out. Could you briefly outline just how outlandish arguments to produce permanents settlements on other planets are in the light of the current climate and ecological crisis?

We have clarified this point by adding: "Recent astronomical observations show that Earth-size, rocky planets are ubiquitous, but this does not mean any of them can support us. What makes Earth hospitable is its multi-billion-year relationship with its biosphere." We keep this explanation brief because of the length constraints. We refer the readers to Nicholson & Haywood 2023 for further details.

5)
"Engaging in advocacy does not diminish public trust in scientists (Kotcher et al., 2017); on the contrary, the public expects climate advocacy from us (Cologna et al., 2021). Scientists are the most trusted professionals alongside doctors (Ipsos, 2022)". It's not clear who you wish to aim this advocacy at. I think the argument you are making is addressed to planetary scientists and ask them to use their knowledge and perspectives to become more involved in public discourse and potentially policy making. If that is the case, then some specific examples on how they could do that? What should people do in response to reading your article?

Our target audience is research scientists across the disciplines we represent (we recognize that ESD readers are primarily planetary scientists). The main goal of this opinion piece is to highlight that there is a connection between the primary research in our fields and planetary

crises, and that the connection takes the form of the unique and powerful perspective these fields offer on the history of the Earth-biosphere system. Advocacy is one possible action that individuals may choose to take, though we are keen not to be prescriptive and have therefore avoided specific advocacy positions. The quoted sentence was primarily a pre-emptive response to a counter-argument we often encounter: that science isn't, or shouldn't be, political. Nonetheless, we appreciate that readers may expect some direction on this issue, and we have appended the following: "Advocacy begins with an acknowledgement that our research is relevant to interpreting planetary crises. We envision a near future in which the next generation of scientists recognizes and embraces their role in communicating knowledge in the political arena. This is a collective endeavour, but for an individual it could be as simple as discussions with colleagues, coursework, outreach, and educational initiatives, or as engaged as working directly with policymakers and politicians."

6)
Reviewer 2 made a number of comments referring to the technosphere. Given the central aim of your argument and its intended audience, I do not think it necessary for you to situate your manuscript within that literature. However, concepts and movements such as the technosphere, transhumanism, and longtermism are relevant to the topic. Given the constraints of this submission's format, I do not think you have sufficient scope to sufficiently address this. I would urge you to consider how your argument could be developed into another longer piece of work that could begin to substantively engage with these specific themes as I think it would be a very fruitful area to explore.

We agree that there are many additional topics to explore, and appreciate the Editor's encouragement. We would like to continue thinking and writing about these themes.